# Current Findings on Gut Microbiota Mediated Immune Modulation against Viral Diseases in Chicken

**DOI:** 10.3390/v11080681

**Published:** 2019-07-25

**Authors:** Muhammad Abaidullah, Shuwei Peng, Muhammad Kamran, Xu Song, Zhongqiong Yin

**Affiliations:** 1Natural Medicine Research Center, College of Veterinary Medicine, Sichuan Agricultural University, Chengdu 611130, China; 2Queensland Alliance for Agriculture and food Innovation, The University of Queensland, Brisbane 4072, Australia

**Keywords:** chicken, gut-microbiota, commensal, pathobionts, *Lactobacillus*, *Bifidobacterium*, *E. coli*, *Shigella*

## Abstract

Chicken gastrointestinal tract is an important site of immune cell development that not only regulates gut microbiota but also maintains extra-intestinal immunity. Recent studies have emphasized the important roles of gut microbiota in shaping immunity against viral diseases in chicken. Microbial diversity and its integrity are the key elements for deriving immunity against invading viral pathogens. Commensal bacteria provide protection against pathogens through direct competition and by the production of antibodies and activation of different cytokines to modulate innate and adaptive immune responses. There are few economically important viral diseases of chicken that perturb the intestinal microbiota diversity. Disruption of microbial homeostasis (dysbiosis) associates with a variety of pathological states, which facilitate the establishment of acute viral infections in chickens. In this review, we summarize the calibrated interactions among the microbiota mediated immune modulation through the production of different interferons (IFNs) ILs, and virus-specific IgA and IgG, and their impact on the severity of viral infections in chickens. Here, it also shows that acute viral infection diminishes commensal bacteria such as *Lactobacillus*, *Bifidobacterium*, Firmicutes, and *Blautia* spp. populations and enhances the colonization of pathobionts, including *E. coli*, *Shigella*, and *Clostridial* spp., in infected chickens.

## 1. Introduction

Beginning from the first moment of birth, every single uncovered surface (for example the skin, mouth, vagina, and gut) in warm-blooded animals becomes step by step colonized by a wide assortment of microorganisms, which are known as the microbiota [1,2,3]. Although we have large data sets, extensive research is still required to understand more about physiological functions and dynamics of the microbiota. It is more important in pathological conditions and when microbiota performs its function in the gut as digestion of nutrients and the main producer of many vitamins [4]. During a lifetime, microbiota evolves with the host in composition with nutrition, probiotics and nutraceuticals such ovotransferrin are the main components that maintain the diversity of gastrointestinal tract (GIT) microbiota [5,6,7]. GIT microbiota has many effects on digestion of nutrients, immunity development, and shielding hosts from pathogens [3,8,9]. Intestinal microbiota affects both local and systemic immune responses [10,11]. Germ-free mouse models are extensively used to study microbiota functions of shaping adaptive and innate immune responses [11,12]. Various bacterial species have been recognized that maintain host homeostasis. For instance, small molecules such as bacterial polysaccharide (PSA) from *Bacteroides fragilis* have given proof that symbiotic microbes and its products communicate with and shape the immune response, particularly in the transformation of CD4+ and Foxp3+ [13]. Segmented filamentous bacteria induce the Th17 cells [14] and *Clostridial cluster XIVa* and *IV* induce the colonic Tregs [15]. Gut health improves the health of the poultry flock by enhancing its performance and regulating T cells in the intestine [14,16]. Chicken intestine is inhabited by a variety of commensal microbiota [9]. Of those, Firmicutes, Proteobacteria, and Bacteroidetes are the most important ones [17]. Bird development is severely affected if the gut microbiota or mucosal barrier of the intestine is disturbed [18]. The GI tract has a very reactive environment and pathogens can disrupt the host and its microflora homeostasis, which is called dysbiosis and leads to mucosal infections [19]. Bacterial dysbiosis has been connected to inflammation and changes in immune functions [20]. Changes in the microbial community affects type I IFNs and inflammatory responses of the host [21,22]. Many diseases disturb the stability of intestinal microflora [23,24,25]. Chickens with dysbiosis are more prone to bacterial infection [26]. Studies reflect connections between gut microbiota and distal organs in regulatory functions like gut–lung, gut–brain, gut–skin, and gut–liver axes, which play an important role in many infectious and chronic diseases [27]. In some studies, it is reported that gut microbiota can regulate the antiviral immune response [28] through metabolites such as short-chain fatty acids (SCFAs). The role of SCFAs is well studied in mouse models that show a reduction in inflammatory symptoms by utilizing SCFAs and T regulatory cell suppression in allergic diseases of airways [29]. Recently, there is growing interest to learn the mechanism involved in gastrointestinal tract (GIT) microbiota and infectious and noninfectious disease interaction. GIT microbiota plays a pivotal role to regulate and induce host responses against various pathogens including viruses [30,31,32], bacteria [33,34,35], and fungi [36]. Trans-kingdom associations of viruses and microbiota suggests important role of microbiota in virus replication, development, and progression [37]. Some of the potential mechanisms involved in gut microbiota mediated immunity to pathogens include those involving pattern-recognition receptors (PRRs) such as Toll-like receptors [30,33,34] and nucleotide-binding oligomerization domain-like receptors [38] that recognize microbial-associated molecular patterns (MAMPs).

In the current study, we focused only on the interaction between gut microbiota and viral infections and their impact on immune regulations in chicken. At present only four viral diseases (Avian influenza, Marek’s, Infectious Bursal Disease (IBD), and Newcastle Disease (ND)) re reported with their connections between gut microbiota and immune modulations.

Avian influenza virus (AIV) is a negative sense single-stranded virus having a segmented genome that causes respiratory illness, gastroenteritis, and diarrhea [39]. There are a number of strains of AI and the H9N2 strain is the biggest threat to public health due to its ability to replicate in mammalian tissue [40,41,42], and previous reassortant isolates of Highly pathogenic avian influenza (HPAI) in humans were shown to carry internal genes from avian H9N2 viruses [40,43,44]. Studies confer that gut microbiota elicit the immune response against the influenza virus and they depicts that gut microbiota regulation is a potential source of treatment for respiratory diseases [28,45]. Due to the high mutation rate of influenza viruses, contemporary lack of a reliable antiviral treatment and consistently effective vaccine emphasize the development of novel management and prevention strategies. The gut–lung axis alliance acts as an important mark for the development of such strategies as it is extensively used in many airway diseases. A correlation among gut microbiota diversity and influenza was observed in mice [30]. It was reported that dysbiosis in chicken gut microbiota resulted in higher cloacal and oropharyngeal shedding of avian influenza H9N2 in chickens, which was also linked with compromised type I interferon (INF) expression [41]. Marek’s disease (MD) is a contagious, globally prevalent viral disease of chicken [46] caused by Marek’s disease virus (MDV) or Gallid alpha herpesvirus 2 [47] that mainly targets lymphoid organs such as spleen, bursa of Fabricius, and thymus, thereby infecting B and T cells [48,49]. MDV causes up to 100% mortality [50,51]. The pathological lesions of MDV include mononuclear infiltration of the gonads, peripheral nerves, various viscera, iris, muscles, and skin. Infected chickens develop CD4+ T-cell tumors in visceral organs and enlarged nerves resulting in paralysis, blindness, and eventually death [52,53,54,55,56,57]. MDV also causes chicken gut microbiota dysbiosis [58].

Infectious bursal disease (IBD) is a viral disease of chicken caused by infectious bursal disease virus (IBDV) [59,60,61,62]. IBDV is a non-enveloped virus that belongs to the genus *Avibirnavirus* and the Birnaviridae family [62,63,64,65,66]. IBDV infection causes immunosuppression, which leads to gut-associated secondary infection, resulting in high mortality rates in chicken [67,68,69]. IBDV causes severe damage to the bursa of Fabricius and affects IgM+ B cell production [61,70].

Newcastle disease virus (NDV) is a contagious disease of poultry caused by highly pathogenic strain avian paramyxovirus type 1 (APMV-1) serotype belonging to the genus *Avulavirus*, subfamily Paramyxovirinae, and family Paramyxoviridae [71,72]. It has been reported that NDV infection causes dysbiosis of gut microbiota of chicken, which increases the severity of disease [73]. Despite these rapid advances, there is little information available on the impact of acute viral infection on the quantity, composition, and kinetics of commensal gut microbiota in the chicken.

## 2. Avian Influenza Virus

Avian influenza virus (AIV) subtype H9N2 has tropism for many tissues, including tissues of GIT and the upper respiratory tract of chicken. AIV enters the body through the mucosa of the respiratory tract and GIT [74]. Recent studies have revealed that commensal gut microbiota play a decisive role in viral pathogenesis to regulate the immune response against influenza virus [28,45]. In contrast, dysbiosis of gut microbiota in chicken elicit the severity of disease [39]. The health and diversity of gut microbiota are key factors to diminish the influenza virus infection [28].

## 3. Commensal Bacteria Elicit Immunity

Commensal intestinal microbiota play a crucial role in the health and disease of the chicken by eliciting an immune response against infection and virus clearance. Different commensal bacteria have their own unique role against viral infection by modulating diverse immune mechanisms as reported in Table 1 [75]. The depletion of these bacteria augment the influenza virus disease course and delay the cloacal and oropharyngeal shedding in H9N2 infected chickens as compared to undepleted groups [41,76]. Type-I IFNs comprised of IFN-α and IFN-β are integral parts of the antiviral innate immune response in virus-infected cells and interrupt the viral life cycle by degradation of virus nucleic acids or inhibition of viral gene expression [77,78,79]. Along with IFNs, IL-22 interactively inhibits intestinal viral infections [80] by impeding GIT tissue degeneration, escalating cell proliferation, and modulating inflammation [81]. The expression level of IFN-α, IFN-β, and IL-22 in antibiotic-treated along with AIV infected chickens was markedly diminished compared to undepleted AIV infected chicks. The expression level of type-I IFNs and IL-22 in the antibiotic-treated group was restored to the undepleted group by *Lactobacillus* and fecal microbial transplantation (FMT) [41,76]. Different bacterial genera in GIT modulate the expressions of different AIV antiviral cytokines. IFN-α, IFN-β, and IL-22 expression were positively correlated with *Collinsella*, *Faecalibacterium*, *Oscillibacter*, *Holdemanella*, *Pseudoflavonifractor*, *Anaerotruncus*, *Butyricoccus*, and *Bifidobacterium* while these were negatively correlated with *Clostridium* cluster-XI, *Escherichia*, and *Shigella* species as shown in Figure 1 [41]. Strong recovery was observed in histomorphological structures and the general architecture of the ileum in AIV infected chickens after fecal microbial transplantation (FMT), and probiotic (PROB) supplementation [41].

## 4. AIV Mediated Dysbiosis in Commensal Microbiota

Type-I INFs and IFN-γ are effective antiviral agents in H2N9 AIV infection [21,22,79,82,83,84] that enhance inflammation and mucosal tissue degeneration, and disrupt the commensal gut microbiota diversity, which leads to an increased pathogenic bacterial population and results in secondary bacterial infections [21,85]. Previously, in H9N2 AIV infected chickens, elevated levels of IFN-γ and IL-17A were observed, which caused the dysbiosis of commensal gut microbiota and decreased the number of lactic acid producing bacteria such as *Lactobacillus*, *Enterococcus*, and *Streptococcus* due to an increased population of pathogenic Proteobacteria [86], comprised of *Salmonella*, *E. coli*, *Klebsiella*, and *Shigella*, which produce inflammation in GIT as described in Table 1 [87]. Similar results were also observed in highly pathogenic influenza virus infected mice and increased production of IFN-γ and IL-17A led to intestinal micro flora dysbiosis [88]. An increased growth of pathobionts including *Vampirovibrio*, *Clostridium* cluster-XIVb, and genus *Ruminococcus* was observed in AIV infected broiler chicks [79]. These pathobionts produce proinflammatory cytokines IL-6 and IL-1B [87]. The mucosal epithelium of GIT plays a basic role in digestion and absorption of nutrients acting as a first line of defense against pathogens and preventing the entry of pathogens into the body of the host [89,90]. Damage of the GIT mucosa promotes the translocation of pathogens into body and causes systemic infection [91,92]. Antimicrobial peptides including mucins (MUC), endogenous trefoil (TFF), and tight junction proteins (Claudins, Occludin, and Zona Occludens (ZO)) inhibit pathogenic microbe infection and keep the permeability of the intestinal mucosa intact [93,94,95,96,97,98,99]. In recent studies, it was reported that in H9N2 AIV infected chickens, the expressions of MUC, TFF, Claudins, Occludin, and ZO were significantly reduced and produced inflammation of mucosal epithelium, which led to secondary bacterial infection due to invasion of *E.coli* as presented in Figure 1 [85,86].

## 5. Infectious Bursal Disease Virus (IBDV)

IBDV is an immunosuppressive disease of poultry [67,69], mainly affecting the primary lymphoid organs comprised of thymus and bursa of Fabricius, as well as gut-associated lymphoid tissues (GALT) [100], which act as the first line of defense against invading pathogens and establish systemic immune responses [101]. The immune system is an important contributor for regulating the microbial composition; likewise, it has been also reported that microbiota shapes the immunity [11]. Immunosuppressive diseases impact the development of the intestinal immunity and microbial composition and consequently modify the gut barrier [3,102]. IBDV causes acute infection between two to five days post inoculation [103], during which peak production of proinflammatory cytokines and IBDV replication has been reported [104], which causes gut microbiota dysbiosis, leading to lower abundance of commensals *Clostridium* XlVa [105]. Previously, it was reported that these commensals induce the colonic T regulatory cells to suppress the production of the proinflammatory cytokines [15,106,107,108]. In IBDV infected birds an increased abundance of sulphur reducing Desulfovibrionaceae was observed and these hydrogen sulfides are toxic to mucosal tissue, which leads to severe inflammation of GIT as described in Table 1 [105,109,110]. *Faecalibacterium* is an important butyrate-producing bacteria in the cecum of the chicken [111]. Higher numbers of *Faecalibacterium* are deleterious for *Campylobacter jejuni* replication since butyrate may inhibit the replication of *Campylobacter jejuni* [112]. Recently, a lower abundance of *Faecalibacterium* and higher fecal shedding of *Campylobacter jejuni* in chickens that were co-infected with IBDV and *Campylobacter jejuni* was observed as compared to those infected only with *Campylobacter jejuni* [113,114]. IgA is the main immunological defense against invading pathogens in the gut and it also regulates the microbial diversity in the intestine [115]. IBDV depletes B cell production in the lamina propria of the intestine and cause enteritis [116], which leads to a decrease in IgA and IgG mediated humoral immunity against *Salmonella typhimurium* and *Campylobacter jejuni*, resulting in increased shedding of these pathogens in feces of IBDV infected chickens as shown in Figure 1 [113,117,118,119]. Contrarily to the pathobionts mediated immunosuppressive effects IBDV, it was also reported that probiotic supplementation in IBDV infected broiler chicks enhanced the body weight gain, feed conversion ratio, and antibody titers, and decreased the morbidity and mortality against IBDV infection [120].

## 6. Marek’s Disease Virus

Marek’s disease (MD) is a contagious and globally prevalent viral disease [46]. Marek’s disease virus (MDV) suppresses the immune system and produces inflammatory neurological syndromes, which lead to paralysis and causes 100% mortality in chickens [49,50,121]. MDV affects the immune system, especially B and T lymphocytes [122]. The pattern of the disease to alter the immune system is probably due to the established link between the microbiome and immune system. MDV pathogenesis causes dysbiosis in chicken gut microbiota, which leads to enrichment of certain pathogenic bacterial genera in the cecum [123]. Ovotransferrin, which is a nutraceutical, has anti MDV properties and it enhances the Firmicutes population and diminishes the Proteobacteria [5,124,125,126]. Members of phylum Firmicutes regulate the inflammation by producing anti-inflammatory cytokines through regulatory T (Treg) cell activation as discussed in Table 1 [127]. MDV during the proliferative phase at 28–35 days of its life cycle [49], reduces the Firmicutes population and provides a favorable environment for the four different opportunistic pathogenic *Lactobacillus* spp. and Proteobacteria colonization due to inflammation of the intestinal mucosa [123,128]. *Blautia* is a Gram-positive staining coccoid or oval-shaped, non-motile bacterium that produces metabolites such as succinate and lactate by the degradation of polysaccharides and provides energy to the host [129,130]. In recent studies, an increased *Faecalibacterium* spp. and *Blautia* spp. colonization in control chicks was reported as compared to their MDV infected counterparts, having more *Streptococcus* spp. [128], and these are opportunistic pathogens that develop septicemia, peritonitis, and endocarditis in chicken [131].

## 7. Newcastle Disease Virus (NDV)

Newcastle disease is a contagious disease of chicken [71], which produces hemorrhages and necrosis of the respiratory tract and the digestive system [132], leading to high morbidity and mortality in chicken [133]. NDV infection induces interferon production [134], which increases the lethality of bacterial endotoxin [135] and causes the disproportion of intestinal microbiota in chickens [73]. *Rhodoplanes* are pathogens that produce febrile conditions and cause local infection [136]. In the cecum of NDV infected chickens, an increased abundance of pathogenic *Rhodoplanes*, *Clostridium*, and *Epulopiscium* was detected, which causes the depletion of *Paenibacillus* and *Enterococcus* [137]. Paenibacillus are commensal bacteria that produce antimicrobial substances against a wide range of microorganisms such as fungi, plant pathogenic bacteria, and anaerobic pathogens including *Clostridium botulinum* as shown in Figure 1 [138,139].

## 8. Conclusions

In conclusion, we have reported here that commensal bacteria including *Lactobacillus*, *Bifidobacterium*, Firmicutes, *Faecalibacterium*, *Blautia* spp., and *Clostridium* XlVa play a key role in viral disease prevention and treatment, through competition, by inhabiting the mucosal surface of GIT. Microbiota helps the host in food digestion to produce SCFAs as an energy source. These SCFAs regulate the different anti-viral immune mechanisms by the production of IFN-α, IFN-β, and T regulatory cells, which stimulate secretions of anti-inflammatory cytokines such as IL-22 and promote the humoral immune response by the production of IgA and IgG antibodies to control the severity of virus infection in chicken. Contrarily, it is also observed that these viral infections cause dysbiosis of intestinal microbiota and enhance gut pathobiont colonization such as Proteobacteria, *Clostridium* cluster XI, *Clostridium* cluster XIVb, *Escherichia*, *Shigella*, *Salmonella*, *Campylobacter jejuni*, *Streptococcus* spp., *Rhodoplanes*, *Vampirovibrio*, Desulfovibrionaceae, and genus *Ruminococcus*. These pathobionts augment the severity of virus infections by suppression of anti-inflammatory cytokines, T regulatory cells, and B lymphocytes immunoglobulins production and enhance proinflammatory cytokines comprised of IL-17A and IFN-γ production. Due to the constant emergence of new viral strains that lead to reduced cross-protection by vaccinations against viral infections, there is, therefore, still a need for the development of a more reliable way to solve this riddle. As we observed, the change of a single bacterial genus can have a direct impact on immune system regulation, and supplementation of these specific commensal probiotics in a specific virus infection could be an alternative to restore the innate and adaptive immune mechanisms and combat these severe economic losses in the poultry industry.

## Figures and Tables

**Figure 1 viruses-11-00681-f001:**
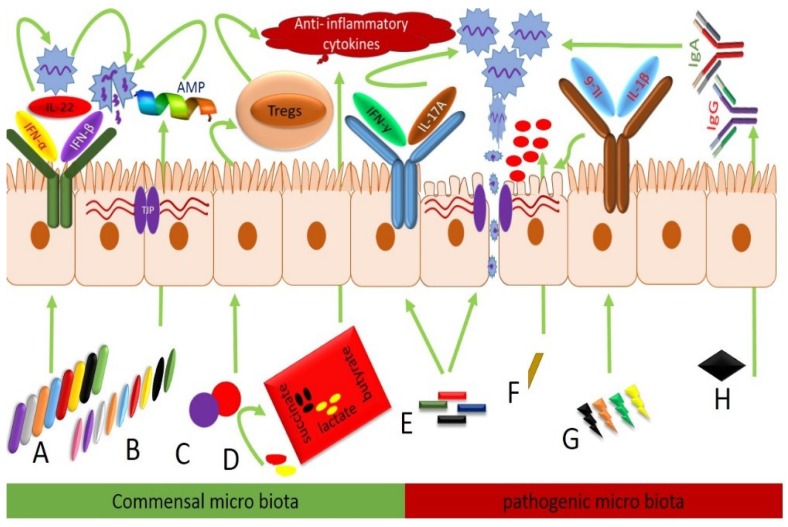
Regulation of different immune mechanisms by intestinal microbiota in AIV, IBDV, MDV, and NDV virus infected broiler chickens. (A) *Collinsella*, *Faecalibacterium*, *Oscillibacter*, *Holdemanella, Pseudoflavonifractor*, *Anaerotruncus*, *Butyricoccus*, and *Bifidobacterium* enhance the IFN-α, IFN-β, and IL-22 secretions, which control the virus replication by degrading the virus nucleus, as well as virus replication genes, and repair mucosal tissue damage. (B) *Bacteroides*, *Candidatus*, SMB53, Parabacteroides, *Lactobacillus*, Paenibacillus, *Enterococcus*, and *Streptococcus* spp. promote the antimicrobial peptides such as MUC, TFF, ZO, and tight junction proteins comprised of claudins, occludin, and zona occludens mRNA expressions and inhibit pathobiont colonization and translocation and suppress inflammation. (C) *Clostridium* XlVa and Firmicutes induce the T regulatory cells, which produce anti-inflammatory cytokines and suppress inflammation. (D) *Faecalibacterium* and *Blautia* spp. enhance butyrate succinate and lactate production, which provide energy and reduce inflammation. (E) Cluster XI, *Salmonella, Escherichia*, and *Shigella* are pathobionts. These pathogens decrease IFN-α, IFN-β, and IL-22 antimicrobial peptides such as MUC, TFF, ZO, and tight junction proteins comprised of claudins, occludin, and zona occludens mRNA expressions, increase the IFN- γ, IL-17A secretions that cause the mucosal inflammation, tissue damage Increased virus replication and fecal shedding. (F) Desulfovibrionaceae produce hydrogen sulfides and produce inflammation of mucosa. (G) *Vampirovibrio*, *Clostridium* cluster XIVb, and genus *Ruminococcus* induce the proinflammatory cytokines IL-6 and IL-1B, which produce GIT inflammation and leads to increased viral replication. (H) *Salmonella typhimurium*, *Campylobacter jejuni* decrease viral specific IgG and IgA production, which results in more viral shedding.

**Table 1 viruses-11-00681-t001:** Comparison between commensal and pathogenic gut-microbiota mediated immune modulation in AIV, IBDV, MDV, and NDV infected chickens.

Virus	Control Group		Infected Group	
	Commensals	Effector Molecules and Outcomes	Pathogens	Effector Molecules and Outcomes
AIV	*Collinsella, Faecalibacterium, Oscillibacter, Holdemanella, Pseudoflavonifractor, Anaerotruncus, Butyricoccus*, and *Bifidobacterium*	Increase IFN-α, IFN-β, and IL-22 and antimicrobial peptides such as MUC, TFF, ZO, and tight junction proteins comprised of claudins, occludin, and zona occludens mRNA expressions	Proteobacteria *Clostridium* cluster XI, *Escherichia, Shigella, Salmonella*, *Vampirovibrio*, *Clostridium* cluster XIVb, and genus *Ruminococcus*	Downregulate the IFN-α, IFN-β, and IL-22 secretion and antimicrobial peptides such as MUC, TFF, ZO, and tight junction proteins comprised of claudins, occludin, and zona occludens mRNA expressions also enhance the secretions of proinflammatory cytokines IFN-γ, IL-17A, IL-6, and IL-1B and produce inflammation
IBDV	Clostridium XlVa	Induce T regulatory cells to produce anti-inflammatory cytokines	Desulfovibrionaceae	Produce hydrogen sulfides and cause inflammation
	*Faecalibacterium*	Enhance butyrate shortchain fatty acids (SCFA) and suppress the inflammation	*Campylobacter jejuni*	Inhibit butyrate SCFA production cause inflammation of GIT
	Probiotics	Increase immunoglobulins, FCR body weight gain	*Salmonella typhimurium* and *Campylobacter jejuni*	Decreased IgG and IgA production
MDV	Firmicutes	Induce T regulatory cells to produce anti-inflammatory cytokines	Pathogenic *Lactobacillus* spp., Proteobacteria	Suppress the T regulatory cells stimulation produce inflammation
	*Blautia* spp. and *Faecalibacterium* spp.	Produce succinate and lactate and provide energy and reduce inflammation	*Streptococcus* spp.	Septicemia, peritonitis, and endocarditis
NDV	*Paenibacillus* and *Enterococcus*	Antimicrobial peptides	*Rhodoplanes, Clostridium*, and *Epulopiscium*	Cause local mucosal infection

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
