# Peer review of "Current Findings on Gut Microbiota Mediated Immune Modulation against Viral Diseases in Chicken"

_viruses, 2019, doi:10.3390/v11080681_

Reviewer 1 Report

Current findings on gut microbiota mediated immune modulation against viral diseases in chicken

Journal: Viruses

Manuscript ID: viruses-540366

Complete List of Authors:

 Muhammad Abaidullah, Shuwei Peng, Muhammad Kamran, Xu Song and Zhongqiong Yin*

Keywords: chicken; gut-microbiota,commensal; pathobionts; Lactobacillus; Bifidobacterium E. coli, Shigella

Comments to Editor and authors

In the manuscript the authors describe the role of commensal bacteria theth provide protection against pathogens (through direct competition, production of antibodies, activation of different cytokines). The review is well detailed and the bibliography is adequate and up to date. 

with regard to nutraceuticals and the protective role of functional foods, such as Ovotransferrin, the authors neglect this aspect by not considering that this protein has numerous protective and immunomodulatory roles both towards the microbiota and/ot the chicken immune system. In my opinion this apsect is crucial  for a correct view of the organism microbiota interaction in infectious diseases

In this regard the authors can refer to the papers below reported: 

Zhu G, Jiang Y, Yao Y, Wu N, Luo J, Hu M, Tu Y, Xu M. Ovotransferrinameliorates the dysbiosis of immunomodulatory function and intestinal microbiota  induced by cyclophosphamide. Food Funct. 2019 Feb 20;10(2):1109-1122. doi: 10.1039/c8fo02312c. PubMed PMID: 30724290.

Giansanti F, Leboffe L, Angelucci F, Antonini G. The Nutraceutical Properties of Ovotransferrin and Its Potential Utilization as a Functional Food. Nutrients. 2015 Nov 4;7(11):9105-15. doi: 10.3390/nu7115453. Review. PubMed PMID: 26556366; PubMed Central PMCID: PMC4663581.

Giansanti F, Giardi MF, Massucci MT, Botti D, Antonini G. Ovotransferrin expression and release by chicken cell lines infected with Marek's disease virus. Biochem Cell Biol. 2007 Feb;85(1):150-5. PubMed PMID: 17464355.

Federica Giardi M, La Torre C, Giansanti F, Botti D. Effects of transferrins and cytokines on nitric oxide production by an avian lymphoblastoid cell line infected with Marek's disease virus. Antiviral Res. 2009 Mar;81(3):248-52. doi:10.1016/j.antiviral.2008.12.008. Epub 2009 Jan 6. PubMed PMID: 19133294.

In the manuscript the authors show an interesting table concerning virus infection (AIV, IBDV, MDV and NDV) and the role of commensal and/or pathogenic microorganisms and the role of some effector molecules. On the next page (line 217), however, there is a caption that recurses to a hypothetical figure 1 absent in the text. The authors must insert the figure that from the caption would seem very interesting and clear.

Minor revisions:

Lane 42: CD4+ and Foxp3+ must be written CD+ < and Foxp3+

Lane 86, 96, 150: bursa of Fabricius should be written in italics

Lane 163, 183, 200: cecum should be written in italics

Lane 250:  lymphocytesimmunoglobulins should be written lymphocytes immunoglobulins

Ref # 15. J., S. The avian immune system. Diseases of Poultry Saif YM, ed Iowa State University Press,Ames. 2003, 5-16. please check that the reference is written correctly according to the journal’s instructions 

For these reasons, after the corrections/additions requested the manuscript can be accepted.

Author Response

Reviewer #1:

In the manuscript the authors describe the role of commensal bacteria theth provide protection against pathogens (through direct competition, production of antibodies, and activation of different cytokines). The review is well detailed and the bibliography is adequate and up to date. 

With regard to nutraceuticals and the protective role of functional foods, such as Ovotransferrin, the authors neglect this aspect by not considering that this protein has numerous protective and immunomodulatory roles both towards the microbiota and/ot the chicken immune system. In my opinion this apsect is crucial  for a correct view of the organism microbiota interaction in infectious diseases

In this regard the authors can refer to the papers below reported: 

•Zhu G, Jiang Y, Yao Y, Wu N, Luo J, Hu M, Tu Y, Xu M. Ovotransferrinameliorates the dysbiosis of immunomodulatory function and intestinal microbiota  induced by cyclophosphamide. Food Funct. 2019 Feb 20;10(2):1109-1122. doi: 10.1039/c8fo02312c. PubMed PMID: 30724290.

•Giansanti F, Leboffe L, Angelucci F, Antonini G. The Nutraceutical Properties of Ovotransferrin and Its Potential Utilization as a Functional Food. Nutrients. 2015 Nov 4;7(11):9105-15. doi: 10.3390/nu7115453. Review. PubMed PMID: 26556366; PubMed Central PMCID: PMC4663581.

•Giansanti F, Giardi MF, Massucci MT, Botti D, Antonini G. Ovotransferrin expression and release by chicken cell lines infected with Marek's disease virus. Biochem Cell Biol. 2007 Feb;85(1):150-5. PubMed PMID: 17464355.

•Federica Giardi M, La Torre C, Giansanti F, Botti D. Effects of transferrins and cytokines on nitric oxide production by an avian lymphoblastoid cell line infected with Marek's disease virus. Antiviral Res. 2009 Mar;81(3):248-52. doi:10.1016/j.antiviral.2008.12.008. Epub 2009 Jan 6. PubMed PMID: 19133294.

Response:

Thank you so much for your careful reading. We cited given articles in our paper. (Line 35-37 page 1, Line 91 and 92 Page 4)

In the manuscript the authors show an interesting table concerning virus infection (AIV, IBDV, MDV and NDV) and the role of commensal and/or pathogenic microorganisms and the role of some effector molecules. On the next page (line 217), however, there is a caption that recurses to a hypothetical figure 1 absent in the text. The authors must insert the figure that from the caption would seem very interesting and clear.

Response:

As per your kind suggestion we have inserted the hypothetical figure at the top of page 6 next to table at page five line 227.

Minor revisions:

Lane 42: CD4+ and Foxp3+ must be written CD+ < and Foxp3+

Lane 86, 96, 150: bursa of Fabricius should be written in italics

Lane 163, 183, 200: cecum should be written in italics

Lane 250:  lymphocytesimmunoglobulins should be written lymphocytes immunoglobulins

Response:

As per your instructions we have made changes in the lanes as given below.

Lane 44 page 1: CD4+ and Foxp3+ must be written CD+ < and Foxp3+

Lane 88 page 2, lane 98 page 3 and lane 158 page 4: bursa of Fabricius to bursa of Fabricius

Lane 171 and  191 page 4,  lane 210 page 5: cecum to cecum

Ref # 15. J., S. The avian immune system. Diseases of Poultry Saif YM, ed Iowa State University Press,Ames. 2003, 5-16. please check that the reference is written correctly according to the journal’s instructions 

Response:

Thank you so much for your careful reading. We have revised the reference as you suggested.

Ref # 18. Sharma, J. The avian immune system. Diseases of Poultry Saif YM, ed Iowa State University Press,Ames. 2003, 5-16.

For these reasons, after the corrections/additions requested the manuscript can be accepted

Reviewer 2 Report

Well written review! I suggest to accept it!

Author Response

well written. I suggest to accept it.

Thank you.

 Reviewer 3 Report

Current findings on gut microbiota mediated immune modulation against viral diseases in chicken

 The authors reviewed recent findings on microbiota changes and their immune modulation during major chicken pathogenic virus infections. This review summarized most of important papers and provided current knowledge and information on patho-immuno-biological roles and functions of chicken microbiota.

 General and specific comments

- Although AIV, NDV and IBDV are important viral pathogens their pathogenicity is diverse from non-pathogenic to highly pathogenic, and is affected by host’s immunity. In case of highly pathogenic AIV, velogenic NDV, and very virulent IBDV they cause acute mortality the role or function of microbiota is of no meaning. Most of the papers referred by this review are dealing with low pathogenic H9N2 AIV, nonpathogenic NDV (more than 120hrs of MDT), and infection of very virulent IBDV with remnant maternal antibody (2-week-old commercial chicks). Therefore, the authors need clearly describe these points.

- The effect of bacterial ghost (BG) on ND vaccine is not related to microbiota, please consider removing it.

 - Many spacing errors, some typos, abbreviation usage and grammatical errors: the manuscript need to be revised as a whole.

Author Response

Comments and Suggestions for Authors

Current findings on gut microbiota mediated immune modulation against viral diseases in chicken

 The authors reviewed recent findings on microbiota changes and their immune modulation during major chicken pathogenic virus infections. This review summarized most of important papers and provided current knowledge and information on patho-immuno-biological roles and functions of chicken microbiota.

 General and specific comments

- Although AIV, NDV and IBDV are important viral pathogens their pathogenicity is diverse from non-pathogenic to highly pathogenic, and is affected by host’s immunity. In case of highly pathogenic AIV, velogenic NDV, and very virulent IBDV they cause acute mortality the role or function of microbiota is of no meaning. Most of the papers referred by this review are dealing with low pathogenic H9N2 AIV, nonpathogenic NDV (more than 120hrs of MDT), and infection of very virulent IBDV with remnant maternal antibody (2-week-old commercial chicks). Therefore, the authors need clearly describe these points.

Response:

Thank you for your careful reading and substantial suggestions. We have revised the text marked in red color. To the best of our knowledge, we have not found any literature regarding gut microbiota mediated immune modulations in case of acute AIV, NDV and Markes disease infections in chicken. However, we have cited reference regarding gut microbiota mediated immune modulation influenza virus infected mice.

Lane 102-104: Despite of these rapid advances, there is little information on the impact of acute viral infection on quantity, composition and kinetics of commensal gut micro biota in the chicken is available.

Lane 142-144: Similar results were also observed in highly pathogenic influenza virus infected mice increased production of IFN- ϒ and IL-17A leads to intestinal micro flora dysbiosis.

-The effect of bacterial ghost (BG) on ND vaccine is not related to microbiota, please consider removing it.

Response:

As per your kind suggestions we have removed the literature related to effect of bacterial ghost (BG) on ND vaccine.

 - Many spacing errors, some typos, abbreviation usage and grammatical errors: the manuscript need to be revised as a whole.

Response:

Thank you so much for your careful reading we have revised the manuscript as you suggested with tracked changes.

Round  2

Reviewer 3 Report

Most of comments were properly answered, and manuscript was revised.